# An Examination of Underlying Domains in Childhood Adversity: A Scoping Review of Studies Conducting Factor Analyses on Adverse Childhood Experiences

**DOI:** 10.3390/ijerph21111441

**Published:** 2024-10-30

**Authors:** Keith Willoughby, Serena Atallah, Kim Arbeau, Jenn Pearce, Thomas Ketelaars, Jeff St. Pierre

**Affiliations:** 1Child and Parent Resource Institute, 600 Sanatorium Road, London, ON N6H 3W7, Canada; serena.atallah@ontario.ca (S.A.); kim.arbeau@ontario.ca (K.A.); jennifer.pearce@ontario.ca (J.P.); thomas.ketelaars@ontario.ca (T.K.); 2Department of Psychology, Social Science Centre, Western University, London, ON N6A 5C2, Canada; jstpierr@uwo.ca

**Keywords:** adverse childhood experiences, negative life events, factor analysis, childhood trauma questionnaire, scoping review

## Abstract

There is an abundance of research linking experiences of childhood adversity to negative physical and mental health outcomes. Areas that remain to be explored and expanded upon include the ideal set of events for inclusion in measures of childhood adversity and testing the models of risk (e.g., cumulative, specificity, dimensional). In the current paper, we performed a scoping review to develop a comprehensive list of studies that conducted factor analyses of childhood adversity measures. There were 89 articles that met the inclusion criteria; trends in the underlying factor structures are reported. Highly associated yet distinct constructs of adversity have demonstrated empirical utility in predicting outcomes in dozens of studies, with consensus that physical abuse, emotional abuse, sexual abuse, physical and emotional neglect, and household dysfunction offer important predictive value to understanding developmental mechanisms of change. We endorse revisions to one commonly used scale that could offer researchers a consistent and psychometrically robust measure of adversity.

## 1. An Examination of the Underlying Domains in Childhood Adversity: A Scoping Review of Studies That Have Conducted Factor Analyses on Adverse Childhood Experiences

It is well-established that childhood adversity predicts future health challenges. The landmark Adverse Childhood Experiences (ACE) study [1] found that adults who endorsed more adverse childhood events (defined as childhood abuse, neglect, parental and household instability, and domestic violence) were at an increased risk of poor health. Subsequent research has replicated these findings, further supporting a dose–response relationship between adversity and health such that the broader the cumulative history of adverse events a child has experienced, the more likely they are to experience challenges with mental (i.e., substance abuse, suicidality, depression) and physical health (i.e., cancer, heart disease, and lung disease) later in life [2,3]. The public health impacts of early life adversity are significant; most Americans will likely experience at least one adverse event before age 18 [4], childhood adversity explains nearly a third of mental health diagnoses emerging in adolescence alone [5], and annual health costs attributable to early life adversity are estimated to be roughly $748 billion in North America [6].

It is therefore no surprise that calls for adversity prevention and intervention methods emerge frequently in the developmental health literature [7,8,9]. Despite the consensus that addressing the impacts of adversity is an important public health issue, disagreement and variation exist regarding methods and models being used to define and understand the impacts of adversity. For example, some debate the extent to which the original ACE score captures all potentially relevant adversities, as well as the appropriateness of using a cumulative ACE score to model the relation between adversity exposure and risk of poor adaptation and negative outcomes [10,11].

### 1.1. Events Assessed

The original ACE study, a basic retrospective survey assessing ten adversities that often co-occur and predict poor health outcomes, offered no empirical or theoretical rationale for item inclusion or exclusion and no clear factor structure [12]. Critics note that many common, potentially significant experiences were omitted [13], especially those that occur outside the home [14]. Follow-up studies have identified additional childhood adversities which may be linked to significant mental and behavioral health outcomes in adulthood [15], including natural disasters [16], racism [17], or exposure to violence outside the home, such as war [18], community violence [19], or bullying [20].

### 1.2. Models of Risk

When it comes to predicting risk, the most common model (corresponding with the ACE study) is the “cumulative” model, which proposes a dose–response relationship between adversity and risk. Cumulative risk totals the number of categories endorsed (yes/no before age 18) and assigns an adversity score; it does not consider the frequency or impact of adversity within any single category [21]. Drawing on McEwen’s (1998) allostatic load model [22], the cumulative risk approach has received considerable support [23,24]. However, this approach has been criticized for failing to consider that certain experiences may impact development through distinct mechanisms depending on the nature of the experience [25,26]. Attempts to investigate the distinct risk of separate single adversities [27] have been criticized for failing to consider the high co-occurrence of adverse events, making it nearly impossible to study the effects of single events in isolation [26,28]. Dimensional models also exist; the dimensional model of adversity and psychopathology (DMAP) [25], for example, draws on experiential learning theories to suggest that the relationship between adversity and risk can be conceptualized along two dimensions of “threat” and “deprivation”—though it is important to note that the DMAP does not suggest that these dimensions capture an exhaustive list of all adversity types. Experiences of “threat” may increase risk for psychopathology through socioemotional mechanisms by teaching children to be vigilant and responsive to signs of danger; in contrast, experiences of “deprivation” may increase risk for psychopathology through cognitive mechanisms by depriving children of expected social and cognitive enrichment necessary for core cognitive skills such as memory, language development, and learning. In contrast to cumulative models, dimensional models suggest that the unique, experience-specific adaptations made to each dimension can be measured distinctly. Further, in contrast to single adversity models, dimensional models acknowledge that certain adversity types may co-occur and therefore do not attempt to measure single adverse events separately. Dimensional models have shown promise in preliminary research [29] but have yet to be tested widely.

Such variation in the experiences and conceptual models used to investigate the nature and impacts of early life adversity has led to the emergence of numerous published instruments for assessing childhood adversity. Oh and colleagues (2018) [30] reviewed measures that could be used in frontline pediatric settings. Their summary primarily recommended clinical screening tools without adequate psychometric validity and reliability. After examining eight systematic reviews of the ACE literature, the American College of Preventive Medicine recently endorsed population-level research into mitigating the consequences of adversity in childhood. Included in their list of recommendations is a call for funding into research validating and generalizing assessment instruments [31]. Finally, Anderson and colleagues (2022) [12] in their recent call to build a formal infrastructure to assess childhood risk in the U.S., note that the Center for Disease Control and American Public Health Association have begun a process to identify the “most impactful core ACEs” that serve as social determinants of health. A synthesis of child adversity measurement approaches with consideration of underlying constructs will support this applied research.

### 1.3. The Current Study

To address this need, we offer a scoping review of reported factor structures of childhood adversity measures. With this review, we have two aims:To provide researchers and policymakers with a list of adversity measurement tools that have undergone factor analysis, along with sample characteristics and reported factor structures, to assist them with selecting a tool that meets their measurement needs. We expect this will be a helpful resource considering the myriad available tools, measurement methods, and conceptual models represented in the current literature.To explore trends in underlying factor structures that may further clarify how the construct of “early life adversity” emerges across studies. We expect this will be useful for those looking to test conceptual models of adversity in future research. It is anticipated our analysis will be complicated by the inconsistent definitions of what constitutes the measurement of adversity, the problem of cross-sample validation in factor analysis, as well as the diversity of research interests discovered (e.g., relating adversity to health outcomes, to economic hardships, to neuroendocrine functioning, etc.).

We have elected to conduct a scoping review due to the nature of these objectives. Scoping reviews allow an alternative methodological approach to systematic reviews, which is systematized, transparent, and replicable, offering a range of review objectives for which a systematic review approach may be inappropriate or impossible [32,33]. Key applications of scoping reviews include: “To clarify key concepts/definitions in the literature; […] [t]o identify key characteristics or factors related to a concept […] [and] to identify and analyze knowledge gaps” [32] (p. 2); which are all main objectives of this review.

Factor analysis is the focal point of the review because it enables the evaluation of measures with a complex set of item variables [34]. When applied to the analysis of instruments, factor analytic techniques can identify a smaller number of underlying factors that group individual item variables and may correspond to broad underlying theoretical constructs [35]. To provide a comprehensive overview of the underlying factor structures of adversity measurement tools, we included articles that used Exploratory or Confirmatory Factor Analysis (EFA, CFA), accounting for findings that both confirmed support for existing adversity measurement models (CFA) and discovered novel models (EFA) that may warrant future investigation.

## 2. Method

### 2.1. Search and Identification of Studies

This research review has been conducted in accordance with the Preferred Reporting Items for Systematic Reviews and Meta-Analyses—Extension for Scoping Reviews (PRISMA-ScR) [33]. Computerized database searches were conducted in PubMed, PsychInfo, Scopus, EMBASE, and CINAHL. Where possible, search statements were constructed using controlled vocabulary; otherwise, Boolean operators were used to combine keywords. Search statements combined the concepts ‘Factor Analysis’ and ‘Adverse Life Events, or Adverse Childhood Experiences’ or ‘Trauma’. For example, the search terms for PubMed were (“factor analysis”[All Fields]) AND ((“adverse childhood experience*”[All Fields]) OR (“adverse life event*”) OR (“trauma”[All Fields])) Limit: (Since 1998), and the terms for Scopus were (TITLE-ABS-KEY (“factor analysis”) AND (TITLE-ABS-KEY (“adverse childhood experience*”) OR TITLE-ABS-KEY (“adverse life event*”) OR TITLE-ABS-KEY (“Trauma”))) Limit: (Since 1998). The search strategy was not limited by the type of outcome. All searches were completed by 16 February 2023.

The discourse on the measurement of the impact of childhood adversity on health was popularized by the 1998 ACE Study [1]; thus, a decision was made to focus on studies published after this date. Therefore, our search was limited to articles published and indexed from January 1998 to February 2023. Backward and forward citation ‘snowballing’ [36] was also employed through the examination of the reference lists of included articles and the identification of articles that have cited included studies.

### 2.2. Inclusion and Exclusion Criteria and Study Selection

Articles were included if they were published in a peer-reviewed journal in English after 1998, included some measure of negative or adverse childhood events, and included at least one factor analysis. Articles focusing only on adversity or negative events in adulthood or not employing a factor analysis [37] were excluded. Theoretical articles discussing childhood adversity in general were excluded.

The titles and abstracts of all articles identified through database searching were divided between a six-person review team. An initial common sample (n = 100) of titles and abstracts was reviewed by every team member to establish interrater reliability (IRR). While initial IRR calculated using Cohen’s Kappa (k) [38] was assessed as acceptable at this stage (k = 0.40–0.65), ongoing group discussion of discrepant decisions led to higher levels of interrater agreement (k = 0.558–0.811) when IRR was calculated again halfway through the review. IRR was calculated a final time at the end of the review (k = 0.786–0.902) to ensure no significant drift in agreement between reviewers. Articles identified as suitable for inclusion after review of the title and abstract were retrieved and examined by the team. For articles examined in full, any disagreements about final inclusion were resolved through group discussion. Variables extracted from the resulting literature included the name of the instrument measuring adversity, the reported factor structure (number of factors reported and name of each factor), sample size, age of sample when adversity was assessed (childhood/adolescence or retrospectively in adulthood), and modality of assessment (caregiver report, self-report).

## 3. Results

An overview of the selection procedure is provided in Figure 1. In brief, 2497 articles were identified through systematic searching and citation chaining. After the examination of titles and abstracts, 2379 clearly did not meet the criteria. Upon retrieval and examination of the 118 remining full-text articles, 89 were determined to meet the inclusion criteria. Thirty-five of these studies examined the Childhood Trauma Questionnaire (CTQ) and Childhood Trauma Questionnaire-Short Form (CTQ-SF) solely; the remaining 54 analyzed a wide array of existing and original adversity measures. Table 1 presents information related to articles examining the CTQ/CTQ-SF, including article name, CTQ variant examined, and factor structure. Table 2 contains information related to all other instruments analyzed in the included articles. Both tables present details about the article, instrument(s), sample size, age when assessed (childhood/adolescence or retrospectively in adulthood), modality of assessment (caregiver report, self-report), and factors identified. The articles reviewed herein included an analysis of 33 distinct childhood adversity measures (translations of measures were counted with the original as one instance). Three studies developed a bespoke ‘Unnamed measure’ for their analysis; a further five created a measure by combining selected items from several existing measures; these are referred to in Table 2 as ‘Unnamed composite measure’.

### 3.1. Study Characteristics

Of the included articles, well over a third (40%) focused on analysis of the CTQ/CTQ-SF. The CTQ (70 items) [40] is a widely used measure designed to capture retrospective reports of child maltreatment with a 28-item version short form (CTQ-SF) [124] frequently used and psychometrically examined [125]. It has been used with children as young as 12 years old and adults [43,50]. It has been translated into many languages and is widely used internationally as a measure of childhood adversity [43,48,50,61,69,72].

Table 3 and Table 4 contain a summary of study characteristics. Most of the studies in this scoping review used retrospective measures of childhood adversity collected in adulthood; a minority collected experiences of childhood adversity through prospective measures using adolescent or child self-report or caregiver reports (see Table 3). Most (36) non-CTQ studies assessed adversity by querying adults retrospectively, with fewer (24) including child and adolescent participants. For studies that included children and adolescents, 14 employed direct self-report assessments, and 9 employed assessments based upon caregiver reports. Overall, sample sizes ranged from 55 to 95,677 (Table 4). Confirmatory statistical analysis techniques were employed slightly more often (34 studies) than exploratory (32 studies), with 12 studies including both techniques in their analyses.

For the CTQ, all but one study that found three- or four-factor models used adult retrospective measures. Of the 25 studies that validated the five-factor model, 8 of them used adolescent self-report measures while the remainder used adult retrospective reports (some using a combination of adolescent and adult retrospective reports). With regards to non-CTQ studies, we note that the three studies which find poverty/financial hardship as a factor loading on its own are all adult retrospective studies, which may represent the challenge of measuring this construct from the self-report of a child or adolescent. Otherwise, there seems to be no pattern of factor structure for age at assessment, which may be related to the diversity of measures used in this sample.

### 3.2. Events Investigated

The most commonly surveyed adverse events impacting health and wellness (see Table 5) are discussed below: sexual abuse, neglect (physical and emotional), physical abuse, emotional abuse, and household dysfunction. Financial hardship/poverty emerged as a relevant event while examining the physical neglect factor. A category for ‘other’ adversities offers an expanded set of adversity types, including exposure to community violence. Results are reported in terms of factors found and not by study, as in some studies, multiple factors related to the same construct were found (e.g., Physical Abuse Factor 1 and Physical Abuse Factor 2) [81]. Finally, a section summarizing tested conceptual models is included to report on studies and assessment tools that may map onto common models of risk in the literature (i.e., cumulative, specificity, dimensional).

### 3.3. Sexual Abuse

Sexual abuse is consistently confirmed as a distinct factor in CTQ/CTQ-SF research (as a 5-item CTQ-SF subscale). Only one included study [69] arrived at a factor structure for the CTQ that did not include it, and this study used a highly abridged (inadequate) six-item CTQ. Sexual abuse item(s) (we note that ACE tools use only a single question for each adversity category) were included as part of the assessment 43 times in non-CTQ research, with sexual abuse grouping as its own factor structure 20 times (including Rape, Indecent Assault, and Sexual Violence factor). For the remaining non-CTQ research, it was a part of a general abuse/dysfunction within the home factor 20 times or a broader factor centered around community-related items 3 times (i.e., ‘community dysfunction’) [14,82,105].

### 3.4. Neglect (Physical and Emotional) and Financial Hardship/Poverty

Physical and emotional child neglect load as separate factors on most CTQ studies (emotional neglect five-item CTQ-SF scale sample “I felt loved”); however, physical neglect (CTQ-SF eight-item scale sample “I didn’t have enough to eat”) has the lowest CTQ/CTQ-SF factor loading and is sometimes subsumed or combined in a three- and four-factor model [69,70,71]. In non-CTQ factor analytic research, physical neglect item(s) were included 36 times, and emotional neglect item(s) 31 times. Amongst these non-CTQ studies, physical neglect loaded onto its own factor only once [53], but an additional three times if needs were not met for financial reasons (e.g., ‘poverty’, ‘family financial problems’) [86,108,123]. Poverty/financial hardship-related indicators were included among broader adversity categories seven times (e.g., ‘overall adversity’, ‘socioeconomic disadvantage/neighborhood safety) [7,82,85,92,101,108,119]. Neglect related to supervision was indicated in four factors [78,80,83,106]. Meinck and colleagues (2021) [106] had a factor that closely resembled physical neglect but included “not enough food/drink as punishment” which is debatably closer to physical or psychological abuse. Emotional neglect loaded onto its own factor seven times in non-CTQ studies, including factors called Emotional Deprivation, Inadequate Emotional Support, Primary Caregiver Lack of Support, and Secondary Caregiver Lack of Support [82,85,115]. In non-CTQ research, physical and emotional neglect were combined into the same factor 13 times. It was not uncommon for physical neglect (14 times) and emotional neglect (9 times) to be included in broader maltreatment factors. Among the neglect items, two factors incorporated experiences that could be considered cognitive neglect (i.e., education and stimulating environment in the home) [104,119].

### 3.5. Physical Abuse

Physical abuse is a robust factor reflected in all included CTQ research (a five-item CTQ-SF subscale, example “People in my family hit me so hard that it left me with bruises or marks”). In research using other measurement tools, physical abuse item(s) were included 50 times in our review sample; however, physical abuse emerged as its own factor only 14 times. For the remaining instances, physical abuse items were part of a broader factor 36 times, grouping with other forms of intrafamilial abuse 34 times or an even broader factor encompassing all maltreatment plus household dysfunction (e.g., substance use, chaos in the home, parental psychopathology) 21 times. Physical abuse occurred alongside extrafamilial violence or threats to physical safety eight times (e.g., bullying, peer victimization, violence or assault within the community, discrimination, exposure to war/migration, accidents)—in most such instances, extrafamilial items are loaded with intrafamilial items. In sum, physical abuse is consistently a distinct factor on the CTQ scale, yet in non-CTQ research, it is loaded onto its own factor fewer times than it is loaded with other factors characterized by other forms of abuse, threats to physical safety, and maltreatment.

### 3.6. Emotional Abuse

Emotional or psychological abuse items (a five-item CTQ-SF subscale, example “People in my family called me things like ‘stupid, lazy, or ugly’”) were most often found to be a distinct factor in analyses of the CTQ/CTQ-SF. In contrast, non-CTQ research studies included item(s) for this construct within a factor analysis 37 times, yet it emerged as its own factor only three times. For the remaining instances when emotional abuse was included in a factor structure, it was repeatedly loaded with emotional neglect (6 times) or with broader forms of maltreatment and household dysfunction (27 times). Emotional abuse was loaded 4 times with extrafamilial or community-level factors, such as perceived discrimination and peer victimization.

### 3.7. Household Dysfunction (Parental Stress and Absence)

Household dysfunction encompasses a number of adversities, including parental separation/divorce, marital violence, parental incarceration, and parental mental illness/substance use. In non-CTQ research, aspects of household dysfunction (i.e., parental distress or a chaotic family life) were included 53 times, making it a consistently well-represented group of adversities in the literature. However, given the lack of consensus on the definition of “household dysfunction” as a construct, it is difficult to summarize the results of the factor analytic literature. As well, household dysfunction is not a factor scale on the CTQ; however, the physical neglect scale includes “My parents were too drunk or high to take care of the family”. A factor tapping parental psychopathology and absence appears essential to understanding the relationships between child adversity and developmental outcomes; however, it requires more research in defining the construct through factor analysis.

### 3.8. Other Notable Adversities

We did not find that other factors, or factors outside the home, loaded consistently. Using our literature search terms, bullying was reported as a distinct factor only once [94], most often being captured as a component of ‘peer victimization’ and grouped with other physical abuse factors or other forms of community violence. Racism and other forms of perceived discrimination were only included as survey items two times [14,118], embedded in a factor structure related to community dysfunction. Several studies included item(s) related to witnessing war/combat [112], being victimized in the context of armed conflict [116], and being forcibly displaced or experiencing immigration instability [90]. Numerous factor solutions [34,78,86,91,105] included factors related to witnessing violence or abuse as opposed to experiencing it directly (most within the community).

### 3.9. Conceptual Models Tested

In line with a multidimensional model of adversity (as opposed to a cumulative structure), most exploratory studies found multifactorial solutions, with only 6 of 32 studies arriving at a unifactorial model. More than 87% of the studies in our sample arrived at a model with at least two, but no more than five factors; the greatest number of factors found was 10 [82]. Investigations (n = 35) using the most common assessment tool, the CTQ/CTQ-SF, tended to support the five-factor model (Emotional Abuse, Physical Abuse, Sexual Abuse, Emotional Neglect, Physical Neglect); the first 25 studies in Table 1 replicated the five proposed subscales for the CTQ. Across other measures (see Table 2), factor structure necessarily differed due to the lack of consistency in adversities assessed, absent items, or items chosen based on the unique health outcomes that were the focus of interest of each researcher. Cross-validation of factor analysis is poor and conceptual modelling inadequate when item pools differ greatly. As an example of this item diversity within our scoping review, one large random sampling community study interested in prospectively surveying adolescent substance use and ACEs chose not to ask about self-reported child abuse victimization due to concerns that they could not ethically follow-up to offer child protection, while adding items regarding contact with child welfare and parental gambling [75]. This demonstrates the wide range of variation that is possible amongst adversity measures.

The original ACE study utilized a single cumulative risk measure and includes only single item sampling of major categories of adversity. Our review indicates unifactor, bifactor (maltreatment, household dysfunction), and other factors based on the many ongoing alterations to the original ACE item pool. A simple two-factor abuse versus neglect model was not readily evident in this factor analytic literature. Three studies [101,115,119] in our review explicitly explored the ‘threat and deprivation’ dimensional model of adversity and psychopathology proposed by McLaughlin and Sheridan (2016) [25].

### 3.10. Positive Childhood Experiences

Positive childhood experiences were captured in some manner in seven of the included measures. Some measures, including the CTQ, incorporate reverse worded items; a common strategy employed in survey design, to encourage participants to read every question carefully [126], and to aid in detection of cases of careless or random response patterns [127]. These items are worded in a manner where higher levels of endorsement correspond to lower levels of experienced adversity (in contrast to the majority of other items for which a higher endorsement would correspond to greater experiences of adversity). On the CTQ, these items reflect more positive experiences or protective factors (e.g., ‘my family gave me strength and support’, ‘I felt loved’). Notably however, a factor combining these items reflecting positive experiences did not emerge in any included factor analysis of the CTQ.

A further six studies in our analysis included items which captured positive experiences in some way. Abbot and Slack (2021) included items from the Childhood Caregiving Environment (CCE) scale (e.g., ‘How often did your family laugh together?’, How often was there an adult in your household who made you feel safe and protected?”’) and found a negative correlation between these items and their included ACE items [73]. Cohen-Cline (2019) included positive items related to ‘Strong and Supportive Relationships’ (e.g., ‘Do you have a family member who made you feel loved important or special) and ‘Health and Wellbeing’ (e.g., ‘Were you generally happy during this time’) [85]. These variables were loaded together in their analysis into a single factor related to support. Gonzalez-Vazquez (2019) included an analysis of the EARLY scale, which includes a 19-item positive experiences subscale which reduced to a single ‘positive family experiences’ factor in their analysis [91]. Langlois and colleagues (2021) included the School Connectedness Scale (e.g., ‘The teachers at your school treated students fairly’) [100]. In their analysis, lack of school connectedness loaded onto a more general factor, ‘Neglect and lack of Connectedness’. Meinck and colleagues (2017) included a measure of life satisfaction (Cantril Ladder Score) as well as a measure of health-related quality of life (KIDSCREEN-10) in their study, finding that health-related quality of life was negatively associated with two of their factors (sexual abuse and physical abuse) [107]. Veredhus and colleagues (2021) also included a measure of subjective quality of life but did not find it impacted their factor analysis [121].

## 4. Discussion

### 4.1. Common Approaches to Early Life Adversity Measurement

In a striking historical shift, the high incident rates of early life adversity and associated detrimental impacts are now increasingly acknowledged as a significant concern by individuals and by societies dealing with healthcare costs [15]. The empirical literature over the past 20 years has frequently surveyed adults in the general population and diverse clinical samples regarding their history of early negative life events, especially regarding child abuse. Less commonly in the literature, children are sampled prospectively on the adversities they experience. Factor analyses of these adversities indicate that multiple, highly associated factors, mostly focused on maltreatment and deprivation within the home, hold distinct statistical relations to a wide variety of health outcomes. We found no clear consensus in this factor analytic literature regarding the core domains that must be included in studying possible sequelae of childhood adversity. Our scoping review indicates there is merit in social health and biomedical researchers examining multiple factors as determinants of adult adaptive outcomes rather than relying on a single child adversity score. The single cumulative rating of the original ACE questionnaire has justifiably received much attention in terms of public health policy but has received little factor analytic support. The CTQ/CTQ-SF most often generates a five-factor model and has received the most empirical attention in terms of factor analyses of the constructs of childhood adversity, yet it lacks key indicators of ‘household dysfunction’ such as parental pathology, distress, and absence that appear frequently in developmental outcomes research.

### 4.2. The Childhood Trauma Questionnaire

The CTQ was overwhelmingly the most common tool investigated in our literature search, with 35 articles that included a factor analysis of the CTQ/CTQ-SF. In general, studies have found five factors when exploratory or confirmatory factor analyses are conducted, although models with three [70,71,72] and four factors [46,66] and even a few for a single global factor [57,71] have been reported. Wright and colleagues (2001) [62] demonstrated that a four-factor model was best for women, and the five-factor model was most suitable for men. Forde and colleagues (2012) [47] suggested that male and female youth may interpret items uniquely. Dudeck et al. (2015) [46] found that a five-factor model fits well for clinical and inmate populations, but a four-factor model was a better fit for a student population. The findings of this scoping review confirm that five factors (Physical Abuse, Emotional Abuse, Sexual Abuse, Emotional Neglect, and Physical Neglect) on this scale hold predictive validity and are well replicated globally across diverse samples of interest (See Table 1). Bernstein and colleagues (2003) [124] found that the five factors are all significantly correlated. Statistically, the weakest factor loadings exist for physical neglect; we note that this scale lacks homogeneity and needs revision. In their recent review, Georgieva et al. (2023) [128] examine and critique five common child maltreatment measures. They find the CTQ-SF to have the largest body of research validation and note in detail psychometric strengths and weaknesses. They note a newer instrument (Maltreatment and Abuse Chronology of Exposure Scale) is promising; however, we could find no factor analytic support of the many subscales in that tool. Both these tools fail to adequately assess the ACE literature variable of “household dysfunction”, which requires definitional clarity yet holds predictive power as a health outcome risk construct. Overall, the results of the scoping review suggest that the CTQ-SF would benefit from the addition of “household dysfunction” items (e.g., domestic violence, parental mental illness, divorce/separation, parental death, parental absence), further strengthening its utility as a valid measure of childhood adversity.

### 4.3. The Adverse Childhood Experiences Survey

The original ten-item retrospective survey helped initiate a new generation of child maltreatment and health risk research [3], which has led to formal calls by many professional organizations (for an extensive review, see Sherin et al., 2022 [31]) for the widespread use of ACE population screening and massive societal changes in funding and implementing preventive health and education policies that could reduce intergenerational toxic stress endured by families and children. The types of multifaceted research needed to translate and implement the science of ACEs into public health benefits have been outlined [7]. Unfortunately, Loveday et al. (2022) [129] note in their extensive literature review that there is little evidence to date that identifying ACE total scores using simple tools at a population screening level has actually led to any improvements in services received by families. Further, it must be noted that in front-line clinical practice, a cumulative single score of risk occurrence is not sufficient, as an understanding of both the nature of the adverse events and evidence of subjective trauma symptoms is needed for patient care [130].

We note that in public health disease prevention efforts, a very small, inexpensive screening of child adversity would be an ideal goal, tempered by a need for demonstrated predictive validity and reliability [131]. At this point, even public health advocates propose expanding the original ACE survey, given concerns with the psychometric validity of the original scale and the maturing literature on conceptual models of childhood adversity [31]. Our review found few researchers using factor analysis endorsed a single adversity score of life events as the best means to understand health outcomes, and an expanded understanding of adversity beyond the ACE scale, with better psychometric and factor validity, is needed by social scientists and neuroendocrine researchers interested in mechanisms of child adaptation.

### 4.4. Retrospective Versus Prospective Measurement

The majority of the studies in this scoping review used retrospective measures of childhood adversity collected in adulthood (CTQ = 32, non-CTQ = 36), with a minority of studies that collected experiences of childhood adversity through prospective measures using adolescent or child self-report (CTQ = 8, non-CTQ = 14) or caregiver report (CTQ = 0, non-CTQ = 9). In our scoping review, there was no indication that factor patterns differed according to time of assessment. Measures of childhood adversity that are assessed retrospectively in adulthood have limitations. A main criticism of collecting this information in adulthood is the reliance on retrospective self-report accounts of events using recall memory, which may not always be accurate [100,101,132]. The age when the event occurred and how long ago it occurred are important considerations [133]. Recall tends to be better in adulthood for childhood events that are serious and are clearly defined (e.g., the death of a parent) [133]. Inaccuracies can occur when adults do not remember an event from their childhood but rely on what they have been told by other people [133]. It has been advised that conclusions based on retrospective reports of childhood adversity be interpreted cautiously [132], especially for events that are subtle/ambiguous and require interpretation [133].

On the other hand, childhood measures of adversity can support early screening, which can help identify at-risk children and youth [30,134]; this also places the researchers in an ethical child protection dilemma to address when seeking research funding and approval. Early screening can support improved recovery due to developmental plasticity in young children [30]. Even when children have not experienced an adverse event, screening can help promote conversations with parents/caregivers and children/youth about safety and the impacts of negative events and relationships on adjustment [7,30]. Notably, ACE scores during childhood may not be stable because additional adverse experiences may be encountered in the remainder of the childhood years [7,90]. Moreover, sometimes parents are the reporters of adverse events during childhood, which may impact accurate reporting due to their lack of knowledge of the event, their own perceptions of the event, social desirability, and/or their preference to not incriminate themselves [30,135,136,137]. Relatedly, a child’s age is important to consider when determining how to collect information about adverse experiences and determining appropriate respondents, with few tools designed for collecting this information in young children (see Oh et al., 2018 for a discussion on this topic) [30].

### 4.5. Conceptual Models

This scoping review demonstrates that for social science researchers, the concept of childhood adversity as a predictor of adult outcomes is best represented by a multifactorial (as opposed to unifactorial) model. A revised CTQ-SF scale could incorporate the five subscales measuring child maltreatment (replicated across broad research in our review) while adding the factor clearly distinguished in other research—parental distress/absence (often called household dysfunction). These types of child maltreatment are expected to be intercorrelated. For example, while emotional abuse is a distinct factor in the many CTQ/CTQ-SF investigations in Table 1, psychological or emotional abuse appears to correlate and co-occur so heavily with the other factors that it may not consistently load as a separate construct on other brief measurement tools. Clinicians link emotional abuse with maladaptive childhood behaviors; however, researchers may have difficulty distinguishing this subtype of emotional abuse from physical and sexual maltreatment. Other adversities were included by investigators in Table 2, and these may be predictive for a specific outcome of interest; however, these events seldom were distinguished in factor analyses across studies. It is recommended that researchers consider routinely adding “Other negative events of significance”—allowing a rater to score stressors that may be essential to their life experience (e.g., war, witnessing community violence, health issues).

The statistical and theoretical research advantages and disadvantages of cumulative risk summary scores in predicting developmental needs have been well examined [21], and recent research has explored the synergistic impact seen when various risk exposures are combined [138]. As noted [25,26], attempting to measure the impacts of specific adversity types may be an unrealistic approach considering the high co-occurrence of adverse events. Sheridan and McLaughlin (2014) [139] propose an alternative dimensional approach to adversity measurement (DMAP); however, only one included study [101] settled on a two-factor model that would indicate the concepts of “threat and deprivation” are sufficient, in part because of the unique contribution of sexual abuse to risk profiles. We recommend future research that attempts to use factor analysis to explore this model further.

### 4.6. Neglect

There is sufficient empirical literature presently to warrant the theoretical distinction of physical/emotional neglect of children as a differential predictor of adult health. We are aware our review failed to sample investigations examining the impact of poverty on children. Low income and low socioeconomic position are often associated with a higher prevalence rate of ACEs [140,141]. Therefore, it is possible that more generalized adversity factors may be found in low-income or low-resource areas in comparison to more specific factors in settings where adversity is less prevalent.

Of the 36 CTQ/CTQ-SF studies, several reported lower internal consistency for the physical neglect factor, indicating that it is unstable and lacks homogeneity, may collapse into a four-factor model, and have suggested that item modifications to this scale are warranted [52,54,64,65,125]. Charak and Koot (2014) [42] discuss nuances of the physical neglect items, speculating financial difficulties may be detected in the physical neglect scale, as opposed to deliberate physical neglect by caregivers. Two non-CTQ studies identified physical neglect and emotional neglect as separate primary factors: Maggiora Vergano et al. (2015) [104] distinguished between “Emotional Neglect” and “Material Neglect” factors, and Lobbestael et al. (2009) [103] found a low intercorrelation between physical and emotional neglect constructs. However, Lobbestael seems to conflate physical neglect with poverty, which may explain this distinction. Overall, while there is some evidence to suggest that physical and emotional neglect may be correlated, the extent of the correlation can vary depending on the specific measures used and the sample being studied. Papers that specifically measured poverty [75,86,108,123] distinguish poverty from neglect in their factor analysis, with neglect and poverty loading on different factors. Indeed, two studies [10,42] explicate that while neglect can be associated with poverty and financial difficulty, poverty alone does not constitute neglect. Many families facing economic hardship provide nurturing and supportive environments for their children, while some families who are not experiencing poverty may experience neglect due to other factors.

### 4.7. Considering Sexual Abuse

Many studies found a factor structure that distinguished sexual abuse from other forms of intrafamilial maltreatment, household dysfunction, and negative community events such as peer victimization. This finding is noteworthy when considering that sexual abuse was distinguished even in cases where other forms of abuse collapsed into higher-level factors [76,84,89]. When seeking to find empirical evidence for McLaughlin and Sheridan’s (2016) DMAP [25], which conceptualizes adversity as a two-dimensional model of “threat” (i.e., abuse) and “deprivation” (i.e., neglect), Sosnowski and colleagues (2023) [115] found that all abuse items *except for* sexual abuse loaded onto one of two higher-level “threat” factors. This finding is particularly interesting because it suggests that sexual abuse may not fit precisely into the threat dimension proposed in the DMAP. As noted earlier, Sexual Abuse is also consistently found to be a distinct factor underlying the CTQ/CTQ-SF. Several investigations have replicated the four-factor structure of another scale, the Early Trauma Inventory–Self-Report (ETI-SR): Physical Abuse, Emotional Abuse, Sexual Abuse, Various Trauma [81]. Because this structure has been replicated with diverse, cross-cultural samples (e.g., Hörberg et al. [93]), this suggests that the experience of sexual abuse may be empirically distinct from other forms of maltreatment in predicting health risks. Putnam et al. (2020) [138] found sexual abuse to be “malignantly synergistic” when combined with other forms of abuse. It is our recommendation that sexual abuse be included in any assessment of early life adversity. Looking ahead, we note that most investigators in our review of this literature did not consider non-physical forms of sexual abuse, such as online exposure to sexual exploitation.

### 4.8. Household Dysfunction

Based on the present literature review, household dysfunction is a distinct factor requiring definitional consensus, incorporating other forms of distress that can, but may not always hinder caregivers from adequately nurturing their child (parental psychopathology and substance use, parental absence and marital dysfunction). We recommend that household dysfunction items be added to the CTQ-SF, moving this scale from one of child maltreatment to a more well-rounded scale of child adversity. While the majority of non-CTQ factor structures variously combined adversities associated with household dysfunction, several authors [10,102,108] specifically found factor structures in which physical separation of children and caregivers (divorce/separation, parental death, parental incarceration) loaded onto a separate factor from other forms of household dysfunction in which caregivers and children remain together with noted stressors (domestic violence, parental mental illness/substance use). Future research may seek to further disentangle which experiences of household dysfunction should be included in an adversity definition and which merely increase the risk for adversity.

## 5. Future Research

### 5.1. Conceptualizing Positive Childhood Experiences

A decade ago, Cicchetti (2013) [142] called upon researchers to explore resilience in young people to understand why we do not see compromised adaptation nor altered biological risks in all children who experience significant maltreatment. It is recommended that future research measurement tools incorporate both risk and protective factors when mapping child development. For example, Anderson [12] proposed building on the ACE (10 item) study with a 16-item scale to be widely utilized in population health research, which adds to the five core maltreatment adversities of physical, emotional, and sexual abuse, emotional and physical neglect. They recommend retaining household dysfunction items of history of parental mental illness, substance use problems, and incarceration while adding two items of risk regarding societal discrimination, one item regarding exposure to community violence, and three items regarding positive childhood experiences (PCE; e.g., “how often were you able to talk to a friend about your feelings) [12]. Our scoping review did not search for factor analytic research into PCEs; however, the addition of PCEs to measures of child development created to predict health outcomes is theoretically intriguing. This resilience construct could be added to a larger item pool (like the CTQ-SF) during Exploratory Factor Analysis, Confirmatory Factor Analysis, and psychometric validation prior to large-scale adoption of any new tool assessing ACEs and PCEs.

### 5.2. Disentangling Neglect and Financial Hardship/Poverty

Financial hardship and poverty can be conflated with deliberate physical neglect when measuring factors related to childhood adversity, as discussed above. We draw researchers’ attention to the risk of misclassification that can lead to recommendations for interventions, policies, and practices that misattribute underlying social determinants of health. Researchers must critically evaluate instruments and their intended purpose at the research development stage and again when seeking approval from research ethics boards, especially for prospective research where child welfare could be involved. Poverty and neglect are unique constructs and should not be used interchangeably; clear definitions and appropriate language are necessary. Future research could examine whether deliberate physical neglect is more accurately associated with abuse when distinct items to detect financial hardship and poverty are used.

## 6. Limitations

The inclusion criteria specified that articles had to be published in 1998 or later, thus omitting any assessments that may have been relevant prior to the publication of the original ACE study [1]. The original 70-item version of the CTQ, for example, was developed prior to 1998 [143]. Studies were excluded if they were not published in English, potentially eliminating psychometrically strong measures of childhood adversity. Our literature search strategy uncovered factor analytic research on child adversity that was focused primarily on intrafamilial abuse, physical and emotional neglect, and household dysfunction. Rarely did exposure to community violence or overall family deprivation appear to load as unique factors. We hypothesize that research interested in broadly assessing the social determinants of health (e.g., surveys of poverty, stress of minority status in a culture, and protective factors (PCE) such as social support [12,31]) may not have been fully exposed by our literature search focusing on ACEs. No analysis was undertaken regarding how the timing (developmental life stage), or chronicity of adversity impacted development. This is largely a methodological limitation, as the questionnaires utilized in our scoping review do not distinguish single-event adversity from chronic adversity, nor the developmental stage.

## 7. Conclusions

Our review indicates that while a cumulative risk score in child development has become a common and valid metric of health outcomes in epidemiological studies, unifactorial single-score summaries of childhood adversity are not common when child maltreatment investigators examine negative life events using factor analytic methods. New measures are now proliferating. The present review indicates the CTQ-SF offers sound psychometric validity of child maltreatment constructs and would be improved with the addition of a domain measuring household dysfunction. As a future direction, consideration should be given to factor analyzing resilience items (PCEs) to enhance the existing ACE research as it relates to developmental outcomes.

## Figures and Tables

**Figure 1 ijerph-21-01441-f001:**
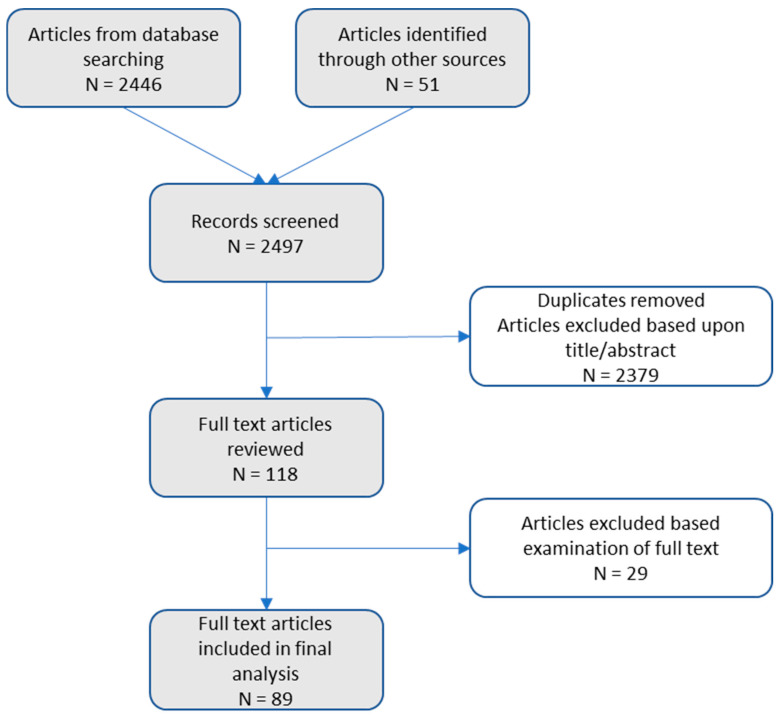
Overview of the systematic literature review.

**Table 1 ijerph-21-01441-t001:** Studies that conducted factor analysis only on the Childhood Trauma Questionnaire.

Article	Measure(s)/PCE	Sample Size, Age, and Modality of Assessment	Factor(s)
Aloba et al. (2020) [39]	CTQ-SF	n = 1337 Adolescent Self-Report	Emotional Abuse Physical Abuse Sexual Abuse Emotional Neglect Physical Neglect
Bernstein et al. (1997) [40]	CTQ (70 item)	n = 398 Adolescent Self-Report	
Bernstein et al. (1998) [41]	CTQ administered CTQ-SF 25 item scored	n = 339 Adult Retrospective	
Charak and Koot (2014) [42]	CTQ-SF	n = 702 Adolescent Self-Report	
Charak et al. (2017) [43]	CTQ-SF (French adaptation, translated to Kirundi. 25 item)	n = 231 Adolescent Self-Report and Adult Retrospective	
Daalder and Bogaerts (2011) [44]	CTQ-SF (Dutch version)	n = 123 Adult Retrospective	
Dovran et al. (2013) [45]	CTQ-SF (Norwegian version)	n = 445 n = 517 Adolescent Self-Report and Adult Retrospective	
Dudeck et al. (2015) [46]	CTQ-SF (German version)	n = 1524 n = 224 Adult Retrospective	
Forde et al. (2012) [47]	CTQ-SF	n = 397 Adolescent Self-Report	
Grassi-Oliveira et al. (2014) [48]	CTQ-SF (Brazilian version)	n = 1925 Adolescent Self-Report and Adult Retrospective	
Güleç et al. (2013) [49]	CTQ-SF	n = 150 Adult Retrospective	
He et al. (2019) [50]	CTQ-SF (Chinese version)	n = 3431 n = 234 Adult Retrospective	
Hernandez et al. (2013) [51]	CTQ-SF (Spanish version)	n = 185 Adult Retrospective	
Karos et al. (2014) [52]	CTQ-SF (German version)	n = 661 Adult Retrospective	
Kongerslev at al., (2019) [53]	CTQ-SF (Danish version)	n = 393 Adolescent Self-Report and Adult Retrospective	
Nakajima et al. (2022) [54]	CTQ-SF (Japanese version)	n = 762 n = 111 Adult Retrospective	
Paivio and Cramer (2004) [55]	CTQ	n = 470 Adult Retrospective	
Petrikova et al. (2021) [56]	CTQ-SF (Slovak version)	n = 1018 Adult Retrospective	
Scher et al. (2001) [57]	CTQ-SF	n = 1007 Adult Retrospective	
Schulz et al. (2014) [58]	CTQ-SF (German version)	n = 2265 Adult Retrospective	
Spinhoven et al. (2014) [59]	CTQ-SF	n = 2308 Adult Retrospective	
Thombs et al. (2007) [60]	CTQ-SF	n = 693 Adult Retrospective	
Thombs et al. (2009) [61]	CTQ-SF (Dutch version)	n = 261 n = 227 Adult Retrospective	
Wright et al. (2001) (Men) [62]	CTQ-SF	n= 916 Adult Retrospective	
Zhang et al. (2013) [63]	CTQ-SF	n = 2090 Adult Retrospective	
Lundgren et al. (2002) [64]	CTQ (Swedish version, 53 item)	n = 55 Adult Retrospective	Emotional/Physical Abuse Sexual Abuse Emotional Neglect Physical Neglect Factor 1 Factor 2
Scher et al. (2001) [57]	CTQ-SF	n = 1007 Adult Retrospective	Emotional and Physical Neglect Sexual Abuse Emotional Abuse and Physical Neglect Physical Abuse
Gerdner and Allgulander (2009) [65]	CTQ-SF (Swedish version)	n = 659 Adult Retrospective	Emotional Abuse Emotional Neglect Physical Abuse Sexual Abuse
Villano et al. (2004) [66]	CTQ-SF	n = 171 Adult Retrospective	
Garrusi and Nakhaee (2009) [67]	CTQ-SF (Persian version)	n = 1000 Adult Retrospective	Emotional/Physical Abuse Sexual Abuse Emotional Neglect Physical Neglect
Lundgren et al. (2002) [64]	CTQ (Swedish version, 53 item)	n = 55 Adult Retrospective	
Wright et al. (2001) (Women) [62]	CTQ-SF	n = 916 Adult Retrospective	
Larsson et al. (2013) [68]	CTQ-SF (Norwegian version)	n = 141 Adult Retrospective	Emotional Abuse/Neglect-Revised Sexual Abuse-Revised Physical Abuse-Revised
Kuttler et al. (2015) [69]	CTQ-SF (six-item short version)	n = 342 Adolescent Self-Report	Physical Abuse Emotional Abuse Emotional Neglect
Innamorati et al. (2016) [70]	CTQ-SF (Italian version)	n = 471 Adult Retrospective	Emotional Neglect/Abuse Sexual Abuse Physical Neglect/Abuse
Hock et al. (2017) [71]	CTQ-SF	n = 139 Adult Retrospective	Emotional/Physical Abuse Sexual Abuse Emotional/Physical Neglect
Spies et al. (2019) [72]	CTQ-SF	n = 314 Adult Retrospective	

**Table 2 ijerph-21-01441-t002:** Studies that conducted factor analysis (excluding studies examining only the Childhood Trauma Questionnaire).

Article	Measure(s)	Sample Size, Age, and Modality of Assessment	Factor(s)	
Abbott and Slack, (2021) [73]	Adverse Childhood Experiences Scale (Abbreviated by authors—8 Items) Childhood Caregiving Environment scale	n = 618 Adult Retrospective	Unnamed Factor 1 Unnamed Factor 2	
Afifi et al. (2017) [74]	Adverse Childhood Experiences Scale (ACE-10)	n = 8316 Adult Retrospective	Physical/Emotional Abuse	
Afifi et al. (2020) [75]	ACE-10 Childhood Trauma Questionnaire Childhood Experiences of Violence Questionnaire	n = 2002 Adolescent (Self and Caregiver Report)	Child Maltreatment and Peer Victimization Household Challenges	
Amini-Tehrani et al. (2021) [76]	Relational Adverse Childhood Experiences Questionnaire	n = 487 Adult Retrospective	Adverse parent/caregiver–child relationship Adverse parent–parent relationship Adverse school relationships Sexual abuse	
Antonopoulou et al. (2017) [77]	Early Trauma Inventory Self-Report Short Form (ETI-SR-SF) (Greek version)	n = 605 Adult Retrospective	Unnamed Factor	
Bailey et al. (2012) [78]	History of Maltreatment and Trauma Form	n = 90 Adult Retrospective and Child (Caregiver Report)	Neglect and emotional maltreatment Sexual abuse Physical abuse Witnessing family violence	
Bethell et al. (2021) [79]	NSCH (National Survey of Children’s Health)-ACEs	n = 95,677 Child (Caregiver Report)	Overall adversity	
Bifulco et al. (2002) [80]	Childhood Experience of Care and Abuse (CECA)	n = 204 Adult Retrospective	Care Control	
Bremner et al. (2007) [81]	Early Trauma Inventory-Self Report (ETI-SR)	n = 288 Adult Retrospective	**ETI** General Trauma Factor 1 General Trauma Factor 2 Physical Abuse Factor 1 Physical Abuse Factor 2 Emotional Abuse Sexual Abuse	**ETI(SF)** General Trauma Random Events Dysfunctional Family Family Accidents Physical Abuse Factor 1 Physical Abuse Factor 2 Emotional Abuse
Brieant et al. (2022) [82]	NSCH (National Survey of Children’s Health)-ACEs	n = 7115 Child (Self and Caregiver Report)	Caregiver psychopathology Socioeconomic disadvantage and neighborhood safety Secondary caregiver lack of support Primary caregiver lack of support Youth report of family conflict Caregiver substance use and separation from biological parent Family anger arguments Family aggression Trauma exposure Lack of supervision	
Brumley et al. (2019) [83]	National Longitudinal Study of Adolescent to Adult Health (6 items)	n = 27,088 Adult Retrospective	Unnamed Factor 1 Unnamed Factor 2	
Chegeni et al. (2020) [84]	Adverse Childhood Experiences Abuse Short Form (ACE-ASF)—Persian version	n = 494 Adult Retrospective	Physical-emotional abuse Sexual abuse	
Choi et al. 2020 [10]	Childhood Experiences Survey (CES)	n = 2413 Adult Retrospective	**ACE-10:** Child Maltreatment Household dysfunction	**CES**: Factor 1 Factor 2 Factor 3 Factor 4
Cohen-Cline et al. (2019) [85]	Unnamed measure	n = 9176 Adult Retrospective	Instability Inadequate emotional support	
Collings et al. (2013) [86]	Developmental Trauma Inventory (DTI)	n = 720 Adolescent (Self Report)	Emotional abuse Domestic neglect Poverty Witnessing domestic violence Witnessing community violence Community assault Indecent assault Rape Domestic assault Domestic injury	
Conway and Lewin, (2022) [87]	ACE-Immigration (ACE-I)	n = 338 Adolescent (Self Report)	Unrest/Violence in Home Country Dangerous Journey Immigration instability	
Cristofaro, (2013) [88]	Trauma Experience Checklist (TEC)	n = 205 Adult Retrospective	Interpersonal abuse and family stress Violence, death and legal involvement	
Ford et al. (2014) [89]	Behavioural Risk Factor Surveillance System (BRFSS) (ACE module)	n = 85,248 Adult Retrospective	Household dysfunction Physical/emotional abuse Sexual abuse	
Giacaman et al. (2007) [90]	Unnamed measure	n = 3415 Adolescent (Self Report)	**Individual:** Home Personal	**Collective:** Strangers Friends/neighbors Tear gas/sound bombs Shelling/shooting/explosions
Gonzalez-Vazquez et al. (2019) [91]	EARLY Scale	n = 522 Adult Retrospective	**Early-FP:** Positive experiences **Early-FN:** Emotional neglect Overprotection Physical abuse Witnessing problems Role reversal High demand Emotional abuse	
Green et al. (2010) [92]	National Co-morbidity Survey-Replication (NCS-R) 12 adversity items	n = 9282 Adult Retrospective	Maladaptive Family functioning Other childhood adversities: Unnamed factor 1 Unnamed factor 2	
Hörberg et al. (2019) [93]	Early Trauma Inventory-SR—Short form (ETISR-SF) Swedish translation	n = 299 Adolescent (Self Report)	General trauma Physical abuse Emotional abuse Sexual abuse	
Iob et al. (2021) [94]	Unnamed composite measure	n = 300 Child (Caregiver Report)	Negative parental experiences Divorce/separation Bullying Abuse	
Ishfaq and Kamal (2020) [95]	Early Trauma Inventory-SR Short Form (ETISR-SF) Urdu translation	n = 479 Adult Retrospective	Physical abuse Emotional abuse Sexual abuse	
Jeon et al. (2012) [96]	Early Trauma Inventory Self Report-Short Form (ETISR-SF) Korean translation	n = 304 Adult Retrospective	General trauma Physical abuse Emotional abuse Sexual abuse	
Karatekin and Hill (2019) [14]	“Expanded ACES” Items from Juvenile Victimization Questionnaire (JVQ) and ACEs scale	n = 4179 Adult Retrospective	Child maltreatment Household dysfunction Community dysfunction Peer dysfunction/property victimization	
Kidman et al. (2019) [97]	ACE International Questionnaire (ACE-IQ)	n = 410 Adolescent (Self Report)	Household disruption Abuse Neglect	
Kogan et al. (2016) [98]	Adverse Childhood Events Scale (11 items)	n = 505 Adult Retrospective	Abusive Parenting Neglect Witnessing Maternal Abuse	
Kristjansson et al. (2016) [99]	Christchurch Trauma Assessment Neglect Scale (Modified)	n = 2594 Adult Retrospective	Childhood physical abuse Childhood sexual abuse Parental partner abuse	
Landis et al. (2003) [34]	Things I Have Seen and Heard Scale	n = 242 Child (Self Report)	**Model 1:** Witnessing Victimization	**Model 2:** Indirect factor Traumatic factor Abuse factor
Langlois et al. (2021) [100]	Original Composite Measure	n = 247 Adult Retrospective	Violence and environmental adversity Interpersonal abuse Neglect and lack of connectedness	
Lian et al. (2022) [101]	Personality and Total Health (PATH) Through Life Project	n = 210 Adult Retrospective	Threat Deprivation	
Liu et al. (2021) [102]	Interview for Traumatic Events in Childhood (ITEC)	n = 2235 Adult Retrospective	Maltreatment Sexual abuse Neglect Divorce Household Dysfunction	
Lobbestael et al. (2009) [103]	Interview for Traumatic Events in Childhood (ITEC)	n = 579 Adult Retrospective	Sexual abuse Physical abuse Emotional abuse Physical neglect Emotional neglect	
Maggiora Vergano et al. (2015) [104]	Complex Trauma Questionnaire (ComplexTQ)	n = 229 Adult Retrospective	**Mother:** Role reversal Physical abuse Psychological abuse/rejection Emotional neglect Failure of protection Material neglect	**Father:** Emotional neglect Failure of protection Material neglect Psychological abuse/rejection Physical abuse
Martin et al. (2013) [105]	Child Exposure to Community Violence Checklist (CECV)	n = 231 Adolescent (Self Report)	Witnessing general violent/criminal acts Directly experiencing and witnessing both family and non-family violence and threats of physical harm Directly experiencing non-family sexual abuse and general feelings of unsafety	
Meinck et al. (2021) [106]	International Society for the Prevention of Child Abuse and Neglect (ISPCAN) Child Abuse Screening Tool Parent Version (ICAST-P)	n = 25,202 Child (Caregiver Report)	General Factor Physical violence Psychological violence Neglect Sexual violence	
Meinck et al. (2017) [107]	Adverse Childhood Experiences—Abuse Short Form (ACE-ASF)	n = 1668 Adolescent (Caregiver Report)	Physical/emotional abuse Sexual abuse	
Mersky et al. (2017) [108]	Childhood Experiences Survey (CES) ACE-10 ACE-17	n = 1241 Adult Retrospective	**10 item model:** Child maltreatment Household dysfunction	**16 item model:** Unnamed factor 1 Unnamed factor 2 Unnamed factor 3 Unnamed factor 4
Morrill et al. (2019) [109]	Unnamed composite measure	n = 1194 Adult Retrospective	Chaotic families Stressful environment Poor family–environment fit	
Osório et al. (2013) [110]	Early Trauma Inventory Self Report (ETISR-SR)—Short Form Portuguese translation	n = 253 Adult Retrospective	General traumas Physical abuse Emotional abuse Sexual abuse	
Ospina et al. (2021) [111]	Alberta ACE survey	n = 1207 Adult Retrospective	Relational Violence Negative home environment Illness at home Sexual abuse	
Saadatmand et al. (2020) [112]	Unnamed measure	n = 440 Adult Retrospective	Having witnessed violence Childhood sexual abuse Personal attacks Witnessing violence amongst adults Direct violence from peers and adults Having witnessed murder	
Schlechter et al. (2021) [113]	Youth and Childhood Adversity Scale (YCAS) Childhood Traumatic Events Scale	n = 1047 Adult Retrospective	Unnamed factor	
Scott et al. (2013) [114]	ACE-10	n = 184 Child and Adolescent (Chart Review)	Abuse Household Dysfunction Mixed	
Sosnowski et al. (2022) [115]	Unnamed composite measure	n = 1662 Adolescent (Self Report)	Threat—non-betrayal Emotional deprivation Sexual assault Threat—Betrayal	
Swingen (2020) [116]	ACE-IQ—Spanish Translation	n = 184 Adult Retrospective	Factor 1 Factor 2 Factor 3	
Thompson et al. (2007) [117]	Things I Have Seen and Heard Scale	n = 784 Child (Self Report)	Community Violence Domestic Violence	
Thurston et al. (2018) [118]	ACE-10	n = 65,680 Child (Caregiver Report) and Adult Retrospective	Household Community	
Usacheva et al. (2022) [119]	Unnamed composite measure	n = 5000 Child (Self and Caregiver Report)	Threat Unpredictability Deprivation	
Vallejo-Medina et al. (2021) [120]	Early Trauma Inventory Self Report-Short Form	n = 2080 Adult Retrospective	General Physical Emotional Sexual	
Vederhus et al. (2021) [121]	Difficult Childhood Questionnaire (DCQ)	n = 28,074 Adult Retrospective	Unnamed Factor	
Weller et al. (2021) [122]	NSCH (National Survey of Children’s Health)-ACEs	n = 1231 Adolescent (Self-Report)	Unnamed Factor	
Wilson et al. (2006) [123]	Unnamed composite measure	n = 235 Adult Retrospective	Emotional neglect Parental intimidation Parental violence Family turmoil Financial need	

**Table 3 ijerph-21-01441-t003:** Study characteristics: age and modality of assessment.

Study Category	Age and Modality of Assessment ^a^					
	Adult Retrospective	Adolescent Self-Report	Adolescent Caregiver Report	Child Caregiver Report	Child Self Report	Child and Adolescent Chart Review
Non-CTQ	36	10	2	7	4	1
CTQ	32	8	0	0	0	0
Total	68	18	2	7	4	1

^a^ some studies had multiple ages and modalities included.

**Table 4 ijerph-21-01441-t004:** Study characteristics: sample size.

Study Category	Sample Size ^a^			
	<100	101–499	500–999	1000+
Non-CTQ	1	22	6	25
CTQ	2	21	7	11
Total	3	43	13	36

^a^ some studies included multiple samples.

**Table 5 ijerph-21-01441-t005:** Events investigated.

Factor	# of Studies Loaded as Its Own Factor		# of Studies Combined into Another Factor	
	CTQ	Non-CTQ	CTQ	Non-CTQ
Sexual Abuse	34	20	0	23
Physical Neglect	29	1 (4 ^a^)	6	27
Emotional Neglect	30	7	5	22
Financial hardship/Poverty	0	3	0	7
Physical Abuse	30	14	7	36
Emotional Abuse	28	3	9	37
Household Dysfunction	0	33	0	20

^a^ Physical neglect was loaded onto its own factor only once, but an additional three times if needs were not met for financial reasons, totaling four.

## Data Availability

The original contributions presented in the study are included in the article, further inquiries can be directed to the corresponding author.

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
