# Peer review of "An Examination of Underlying Domains in Childhood Adversity: A Scoping Review of Studies Conducting Factor Analyses on Adverse Childhood Experiences"

_ijerph, 2024, doi:10.3390/ijerph21111441_

Round 1
Reviewer 1 Report
Comments and Suggestions for Authors
Minor comments
· Measure/s and sample size, age and modality included in Table 2 are not shown in Table 1; it would be useful to keep the tables consistent for comparison – even if ‘measure’ in Table 1 is standard across all studies
· Line 229 - I anticipate that most of the literature, but not all, is related to health and wellness if outcomes were being assessed, but it would useful to clarify in the methods that the search strategy was not limited by type of outcome (since line 229 specifically mentions ‘health and wellness’ and other parts of the paper refer to health outcomes)
· Suggest that the results on the factors found – sections 3.3 – 3.8 be summarized in a table for ease of reference
This paper is well thought out, systematic in its approach, well-written, and is a valuable contribution to the field. I have minor comments/suggestions that could enhance reader experience. My main concern/point of interest is that, given the authors well-informed review of literature and theoretical underpinnings of the field, there are issues that the authors do not raise in their discussion that could be briefly discussed (either interpreted from their results, or raised as points of interest for future research, or addressed as a limitation of the study) that are critical in how we assess ACEs and interpret their influence on outcomes:
1. How the timing of adversity can affect the structure of factors, depending on developmental sensitivity – might there be early trauma factors that are distinct from later adolescent adverse events. There might be varying effects on outcomes depending on timing – e.g. direct abuse in the early years could impact on socio-emotional development but the range of household dysfunction (that the authors do reference as an important but understudied domain) may influence levels of risk taking in adolescence.
2. The prevalence of ACEs in a specific context. In low-resource, lower-income settings where ACES are pervasive across the majority of the population, with more likelihood of co-occurrence, is it possible that this may result in broader, more generalized factors that capture the overlap of adversity compared to more distinct factors in settings where ACEs are less pervasive
3. Chronicity – related to more pervasive ACEs settings but not exclusively, can factor analysis result in factors that reflect long-term chronic stressors like chronic deprivation compared to acute stressors in settings where ACEs are less pervasive and may find more distinct narrower factors
Reviewer 2 Report
Comments and Suggestions for Authors
Thank you for the interesting work. Please see my comments and suggestions below:
- Please change the formatting for the tables to make them more reader-friendly (e.g. Table 2 is occasionally split in the middle of the page with the title (ll.196-198, 200-201, p. 9, 10, 11, 12, 13 etc)
- Creating a table with study characteristics could be helpful (ll 214-217) as well as for family dysfunctions (3.7) and other notable adversities (3.8)
- Did you find any differences in factor loading for retrospective vs prospective questionnaires/studies? Adults vs adolescents? Please articulate your findings in the discussion section
- I’d suggest to create separate result tables for CTQ and ACEs studies instead of one Table 2 to make it easier for the readers
- Please expand and add more arguments to support your conclusion “…however, our review found few researchers endorsed a single adversity score of life events as the best means to understand health outcomes. Psychometrically, a 10-item scale appears insufficient to allow for an adequate understanding of the domains of developmental adversity”. (ll 392-394)
- Considering ongoing debates on the role of poverty, I’d suggest creating a separate theme on poverty and reflecting it in your result tables, and expanding your discussion on it (ll 445 -450).
- Even though you did not search specifically for PCEs in your review, please indicate if any of the studies included in your review mentioned any PCEs as you discussed them in your discussion section (ll.488-506)

Round 2
Reviewer 2 Report
Comments and Suggestions for Authors
Thank you for considering my comments and suggestions